# Self-Esteem, Social Problem Solving and Intimate Partner Violence Victimization in Emerging Adulthood

**DOI:** 10.3390/bs13040327

**Published:** 2023-04-12

**Authors:** Chloé Cherrier, Robert Courtois, Emmanuel Rusch, Catherine Potard

**Affiliations:** 1EE 1901 QualiPsy, Department of Psychology, University of Tours, 37041 Tours, France; robert.courtois@univ-tours.fr; 2EA 7505 EES, Department of Public Health, University of Tours, 37044 Tours, France; emmanuel.rusch@univ-tours.fr; 3UR 4638 LPPL, Department of Psychology, University of Angers, 49045 Angers, France; catherine.potard@univ-angers.fr

**Keywords:** intimate partner violence, emerging adulthood, self-esteem, social problem solving, victimization

## Abstract

Although there are many studies examining the psychosocial vulnerability factors of intimate partner violence (IPV) victimization in emerging adulthood, little is known about the life skills that may be involved, such as social problem solving (SPS) and self-esteem. The aim of the current study is to explore the relationships between SPS, self-esteem, and types (i.e., psychological, physical and sexual) and severity of IPV victimization in emerging adulthood. Based on a French online survey, 929 emerging adults (84.6% of whom were women with a mean age of 23.6) completed self-report questionnaires related to SPS (problem orientations and problem-solving styles), self-esteem and IPV victimization. The results showed that positive SPS skills and higher self-esteem were associated with lower severity of IPV. Multivariate analyses showed that the most associated factors of severe forms of IPV were avoidant and impulsive/carelessness styles. Minor sexual violence was positively associated with lower self-esteem and rational problem-solving skills, while minor psychological victimization was related to avoidant style. Upon completion of this study, it can be said that conflicts which escalate into IPV may be associated with dysfunctional conflict resolution styles, highlighting the importance of interventions that promote the development of life skills in order to prevent IPV.

## 1. Introduction

Maintaining positive, non-coercive romantic relationships is crucial for the well-being of emerging adults and for preventing the onset of violence in adulthood. This is particularly true for emerging adults, who are involved in romantic relationships that contribute to the construction of their identity [1,2,3]. The period of emerging adulthood (i.e., late adolescence and early adulthood, 18–30 years) [4] is characterized by five distinctive feature:, (i) identity exploration: the emerging adult tries to answer the question “Who am I?” and explores several options, (ii) instability, which could be within interpersonal relationships or different environments (work, home, etc.), (iii) self-focus, where the emerging adult makes his own decisions and learns to be self-sufficient, (iv) feeling in-between a transition period between adolescence and adulthood and (v) possibilities/optimism, having the hope that anything is possible and anything can be accomplished [1,4]. At this developmental stage, individuals try out different possibilities in various areas of life, especially in love and relationships. The discovery of oneself and of others and the new emotions that this can arouse can sometimes make it difficult to maintain harmonious relationships, and conflicts sometimes emerge, which can escalate to episodes of intimate partner violence (IPV) [5].

IPV is a worldwide public health issue and is defined as “behaviour within an intimate relationship that causes physical, sexual or psychological harm, including acts of physical aggression, sexual coercion, psychological abuse and controlling behaviors” [6] (p. 11). Therefore, it refers to the violence committed by a partner or ex-partner in the context of an intimate relationship. IPV manifests itself in three main forms: psychological, physical and sexual. Psychological violence refers to non-physical aggressive behaviours in intimate relationships with the aim of harming a partner’s psychological well-being [7]. It involves acts of emotional abuse, manipulation, control, devaluation, isolation or harassment [7]. Physical violence involves the intentional use of physical force with damages ranging from injury to death [8]. Such behaviours can include blows (slaps, punches, etc.) or more generally any use of force against the other: grabbing by the arm, making sudden movements, blocking the way, etc. Sexual violence is divided into three categories: (i) use of physical force to compel a person to engage in a sexual act against his or her will, whether or not the act is completed, (ii) an attempted or completed sexual act involving a person who is unable to understand the nature or condition of the act, to decline participation, or to communicate unwillingness to engage in the sexual act (e.g., due to illness, disability, or the influence of alcohol or other drugs, or due to intimidation or pressure), (iii) abusive sexual contact [8].

The prevalence rates of IPV in the general population can vary widely in research, as shown in a multicentre European study [9]. It highlights that the prevalence rates in the past year of victimization of psychological violence for women ranged from 46.4% to 70.5%, and for men from 48.8% to 71.8%; for physical violence this ranged from 8.5% to 23.1% for women and from 9.7% to 31.2% for men; and for sexual violence, the rates ranged from 8.9% to 25.3% for women, and from 5.4% to 25.3% for men. Nevertheless, emerging adults are particularly affected by IPV since it has been estimated that 45.2% of young women and 40.8% of young men first experience some form of IPV between 18–24 years old [10]. Although IPV is generally described as a gendered act (i.e., perpetrated by men towards women) [11,12], the current literature shows that the prevalence of IPV victimization could actually be more equal and symmetric between women and men than expected, especially during emerging adulthood [13,14,15,16]. While men and women may experience similar levels of physical and psychological IPV, women remain the main victims of sexual violence for those who are in romantic relationships [14,17]. It is more likely also that women will experience more severe injuries, indicating a gender asymmetry in the consequences [18]. The negative consequences of IPV victimization upon health have been well-established [6], including depression, stress, physical injury, alcohol use and socioeconomic issues, as well as the risk of re-victimization [19].

Consequently, over the past decade, researchers have examined how intrinsic vulnerabilities or protective factors may impact involvement in IPV in order to improve prevention and counselling strategies [20]. Furthermore, we know that there are interconnections between victims and perpetrators as well as co-occurrence. Understanding the dynamics of IPV victimization requires recognition that victimization can also be linked to perpetration. The identification of the vulnerability factors of IPV victimization ultimately makes it possible in time to prevent its perpetration, without blaming the victims [21]. A wide range of vulnerability factors for IPV has been clearly identified on different levels. Factors that have been examined thus far relate to individual experiences, such as having witnessed parental violence or having been subjected to child abuse, cognitive and behavioural factors, such as acceptance of violence or alcohol use, as well as relationship factors, such as difficulties in relationships with peers and/or parents [22,23]. In contrast, few studies have focused on factors or abilities that may be protective. Identifying these protective factors would make it possible to establish a positive approach to promote evidence-based prevention of IPV [24,25]. Specifically, improving emerging adults’ life skills could be key to both reducing problematic interpersonal relationships with partners and to preventing IPV.

Life skills are essentially abilities that promote mental well-being and positive social relations. More precisely, they are defined as “abilities for adaptive and positive behaviour that enable individuals to deal effectively with the demands and challenges of everyday life” [26] (p. 3). The 2001 WHO classification [24] divides them into three categories: emotional, social and cognitive. These three types are mutually dependent and are interrelated. While they commonly form the basis of intervention programs, they have been little investigated or mentioned in empirical research. Life skills are closely intertwined with psychological concepts, including self-esteem [24,25]. In the present study, we focused primarily on cognitive skills (i.e., social problem solving), based on the view that IPV can result from an inability to solve problems with one’s romantic partner. Moreover, cognitive skills have been proven to be positively related to constructive, satisfying and long-term relationships [17,27].

### 1.1. Social Problem Solving and Intimate Partner Violence

The development of conflict resolution skills mainly depends on the family context in which the young adult was raised. Theories of parental attachment and social learning will, among other things, impact the acquisition of these skills. An insecure family environment where the young person may witness unconstructive conflict management strategies will tend to negatively impact the development of their conflict resolution skills [24,28,29]. This development continues into the developmental periods of adolescence and emerging adulthood, being influenced, on the one hand, by the cognitive abilities that the young person obtains and, on the other hand, depending on the interpersonal relationships that the young person fosters. The cognitive problem solving theory postulates that teaching social problem-solving skills can improve interpersonal relationships and impulse control, promote self-protecting and mutually beneficial solutions among peers, and reduce or prevent negative “health-compromising” behaviours [26].

Social problem solving (SPS) refers to the ability to solve problems which happen in an individual’s natural environment [30]. These include all types of problems that might affect a person’s functioning, such as intrapersonal problems (e.g., emotional and behavioural problems) or interpersonal problems (e.g., relationship conflicts). More recently, authors have defined it as “the self-directed cognitive-behavioural process by which an individual, couple, or group attempts to identify or discover effective solutions for specific problems encountered in everyday living” [31] (p. 199). It has been shown to be largely determined by two general processes: (1) problem orientation, which is a cognitive-emotional process, and (2) problem-solving style, namely, the skills required to understand problems and find effective solutions [30,32]. For example, faced with a dispute in a romantic relationship, SPS is a cognitive-behavioural process that makes it possible for one to deal with the problem in an attempt, depending on the objectives targeted, to reduce the distress evoked by generating solutions and increasing the probability of choosing the one judged to be the most appropriate. Several articles have investigated the relationship between IPV and general problem solving in emerging adulthood, demonstrating that good problem-solving skills can prevent or reduce the occurrence of conflict in romantic relationships or IPV victimization/perpetration [33,34,35]. More specifically, Bonache et al., in one study on adults and one on adolescents, show that self-reported conflict engagement or withdrawal are positively associated with victimization of physical, psychological and sexual violence, with no gender-related difference (except the link between physical violence and avoidance which was not significant for women) [34,36]. However, the limited number of studies and their heterogeneity in terms of population, tools and conceptualization of problem-solving skills makes it difficult to generalize the results. One study investigated the relationship between SPS and IPV and found that negative problem-solving styles were associated with IPV victimization in adult women [37]. However, to our knowledge, no studies have yet investigated these relationships in emerging adults, although a review of the intervention research literature indicates that interventions focused on problem solving for victims of IPV are promising for maintaining mental and physical well-being [38].

### 1.2. Self-Esteem, Social Problem Solving and Intimate Partner Violence

Self-esteem refers to a general perception of one’s own value that builds up over time [39]. It increases from childhood to adolescence and reaches its peak in emerging adulthood. Women report lower levels of self-esteem than men, but the course of development converges later in life [40]. In the same way as conflict resolution, self-esteem is built according to life experiences related to the social environment [41]. Psychosocial adversity in emerging adulthood can lead to lower self-confidence and can negatively impact self-esteem. Thus, in the event of conflicts within romantic relationships, emerging adults can find it more difficult to activate their life skills, which in turn can make them more vulnerable to violence [24,42]. This is why self-esteem has been identified as a protective (high self-esteem) or an at-risk (low self-esteem) factor for IPV victimization in emerging adults [22,23,43], for men [44] and women [45]. It can indeed be the consequence of IPV victimization (mainly studied in women) [43,46,47,48,49] or involved bidirectionally [45]. Few papers have studied the relationships between SPS, self-esteem in romantic relationships and IPV [48,50,51,52]. Although SPS and self-esteem have been found to be significantly related [32,53], research has not yet examined how these two variables are associated with the three forms of IPV victimization in emerging adulthood. For example, D’Zurilla in 2003 [54] examined these links for aggression and suggested that low self-esteem and deficits in problem-solving ability may be important risk factors for violence.

### 1.3. The Current Study

Studies on IPV victimization in Europe are still poorly documented and focus either on adolescents [55] or on women only [56]. Nevertheless, as has been shown, emerging adulthood is an important period of life for developing long-lasting romantic relationships. Furthermore, the gender symmetry of IPV leads to the inclusion of men and women in the studies. This French study therefore aims to continue European research to have a clearer understanding of IPV in order to better prevent it by attempting to bring theoretical concepts of psychology (i.e., SPS and self-esteem) closer to a concept of public health (i.e., life skills), currently poorly documented in empirical research. As IPV victimization and SPS in emerging adults have not yet been the focus of much research, this study explores the relationships between SPS, self-esteem and the type and severity of current IPV victimization (psychological, physical and sexual violence), and investigates which are most involved. The paucity of research on these topics suggests the need for a specific focus on all SPS sub-dimensions. Moreover, given that childhood abuse, as well as certain socio-demographic factors, e.g., gender, age and level of education, can be vulnerability factors for victimization of IPV [22,23], they were investigated and controlled in the study. Childhood abuse can explain the lack of SPS skills and impact upon self-esteem, making one more vulnerable to psychological or physical violence victimization in emerging adulthood [37,57,58]. Some emerging adults may avoid interpersonal relationships due to childhood abuse, which may adversely impact their ability to solve problems effectively in social situations and increase the risk of IPV [37]. We firstly hypothesize that greater severity of psychological, physical and sexual violence will relate to lower self-esteem and poor SPS skills. Secondly, we hypothesize that negative SPS skills (i.e., negative problem orientation, impulsive/carelessness style, avoidance style) as well as low self-esteem, will be linked to a greater risk of becoming a victim of the three forms of IPV.

## 2. Materials and Methods

### 2.1. Participants and Procedure

A total of 929 emerging adults in France (*M*_age_ = 23.61 years, *SD* = 3.36, range = 18–30 years), including 786 women (84.6%) and 143 men (15.4%) participated in this study. They included 430 (46.3%) full-time students, 139 (15%) students in part-time employment, 272 (29.3%) workers, and 88 (9.5%) unemployed. The mean academic level was 15.04 years (*SD* = 2.72). The design of the study was cross-sectional, and was carried out in 2019, from April to August. A self-administered online questionnaire designed for young adults was distributed via various networks in France to target a general population (i.e., social networks, emailing students in the university, emailing public health actors, etc.). Participants gave their consent, anonymity was guaranteed and no financial compensation was awarded. The project was approved by a Research Ethics Committee of Tours-Poitiers (2019-03-04).

### 2.2. Measures

#### 2.2.1. Socio-Demographic Variables

Participants completed a demographic information section that included questions on gender, age, education level and history of childhood abuse (adaptation of the 5 dimensions of the French version of the Childhood Trauma Questionnaire) [59].

#### 2.2.2. Intimate Partner Violence

The French version of the revised Conflict Tactics Scales (CTS2) [60] was used to assess psychological, physical, and sexual victimization (39 items). Participants reported the frequency of each tactic within the past year on a Likert-type scale ranging from 0 (never) to 6 (more than 20 times). Each dimension of the tool can be subdivided into minor or severe acts of violence, indicating the severity of the assault [61]. For example, for minor psychological violence, one of the items is “my partner insulted me or swore at me”, and for severe acts, “my partner destroyed something belonging to me”. For minor physical violence, one of the items is “my partner pushed or shoved me”, and for severe acts, “my partner choked me”. For minor sexual violence, one of the items is “my partner insisted on having sex with me when I did not want to (but did not use physical force)”, and for severe acts, “my partner used threats to make me have oral or anal sex”. The CTS2 scale therefore makes it possible to measure the levels of severity of the three forms of IPV by categorizing them into three mutually exclusive types: absence, minor only and severe. Items categorized as severe violence have a higher potential for injury, which is what differentiates them from those found on the minor violence subscales. It should nevertheless be noted that Straus and Douglas [61] point out that the term “minor” should not be interpreted as suggesting something that is not a serious problem for either the victims or society. In addition, victims who have experienced a severe act have often also experienced a minor one. The past-year prevalence was assessed as the rate of participants that reported having been a victim of at least one act of violence during the 12 months leading up to this study. In this study, Cronbach’s alphas range from 0.60 to 0.88. CTS2 is one of the most widely used measures to assess IPV amongst emerging adult populations [62], especially internationally [23] or in France [63].

#### 2.2.3. Self-Esteem

The French version of the Rosenberg Self-Esteem scale (RSE) [64] has 10 items evaluating global self-esteem. The questionnaire is scored on a 4-point Likert-type scale ranging from 1 (strongly disagree) to 4 (strongly agree). Higher scores reflect more positive self-esteem. In this study, internal consistency for the RSE is adequate (α = 0.91).

#### 2.2.4. Social Problem Solving

The short French version of Social Problem-Solving Inventory-Revised Short-Form (SSI-R: SF) [32] is used. This is a self-report questionnaire measuring SPS skills in the individual’s affective, cognitive and behavioural responses to real-life problem-solving situations. Twenty-four items evaluate the five SPS subscales, with two relating to problem orientation, positive problem orientation (PPO; i.e., “I try to see my problems as challenges”) and negative problem orientation (NPO; i.e., “I feel afraid when I have important problems”), and three concerning problem-solving style: rational problem solving (RPS; i.e., “When solving problems, I think of many different options”), impulsive/carelessness style (ICS; i.e., “When solving problems, I go with the first good idea that comes to mind”) and avoidance style (AS; i.e., “I wait to see if a problem goes away before trying to solve it myself”). The items are rated on a 5-point Likert scale ranging from 1 (not at all true) to 5 (extremely true). Higher scores on total SPSI-R:SF, PPO, and RPS subscales indicate good SPS skills; whereas higher scores on the NPO, ICS, and AS subscales indicate maladaptive SPS skills. Internal reliability coefficients in the present study range from 0.67 to 0.85. 

### 2.3. Data Analysis

The severity of IPV victimization of emerging adults was categorized into three groups (absence, minor only or severe) for each form of violence, psychological, physical and sexual, according to the recommendations of Straus and Douglas [61]. The normality of the data was calculated by skewness and kurtosis for each variable. We conducted descriptive analyses of participants’ socio-demographic and IPV characteristics, using means, standard deviations and percentages of the variables. Chi-square tests were conducted to observe whether the severity of IPV differed by gender. Fischer ANOVAs were conducted to test the differences in socio-demographic variables, and for self-esteem, and mean SPS scores were calculated between groups of type and severity of IPV victimization. Post-hoc comparisons were evaluated using the Bonferroni test. We also conducted Pearson correlations between self-esteem and SPS. Finally, multinomial logistic regressions were performed to determine the variables most associated with the severity of IPV (using absence of violence as the control group). We report odds ratios (OR) with 95% confidence intervals. Each of the regression models was adjusted for age, education level and history of childhood abuse. Interactions between self-esteem, SPS and gender were tested without yielding significant results. Statistical analyses were conducted using SPSS version 25.

## 3. Results

### 3.1. Descriptive Statistics

Socio-demographic characteristics (gender, age, education level and history of childhood abuse) and severity of IPV (psychological, physical or sexual) experienced by participants are shown in Table 1 and Table 2. The prevalence of having experienced IPV at least once is 64.0% (*n* = 595) for psychological violence, 16.8% (*n* = 156) for physical violence, and 20.5% (*n* = 190) for sexual violence. Moreover, 16.1% (*n* = 150) of participants had already experienced psychological and physical violence, 17.1% (*n* = 159) psychological and sexual violence, 6.6% (*n* = 61) physical and sexual violence, and 6.5% (*n* = 60) had experienced all three forms. Only 3.2% (*n* = 30) have never experienced any of the three forms of IPV.

Regarding gender differences, more than half of the women and the men reported minor or severe psychological violence (no gender difference; *χ*^2^ = 0.57, *p* = 0.75), an average of 13.4% (*n* = 116) of women and men had experienced minor physical violence and 4.0% (*n* = 40) severe physical violence (no gender difference; *χ*^2^ = 0.95, *p* = 0.62), and there were more female victims of sexual violence (22.4 %, *n* = 176; including minor sexual violence: 21.1 %, *n* = 166) than males (9.8 %, *n* = 14; *χ*^2^ = 12.25, *p* < 0.05). Only women had been victims of severe sexual violence.

Table 2 presents the other descriptive information, indicating that older participants experienced minor psychological violence; a low education level was associated with severe physical violence; and participants who had experienced more childhood abuse experienced more minor to severe psychological, physical and sexual violence.

### 3.2. Comparisons between Self-Esteem, Social Problem Solving and Severity of Intimate Partner Violence

To investigate in greater depth which variables are the most associated with the severity of the three forms of IPV victimization, we carried out multinomial logistic regressions to determine whether self-esteem and SPS skills were more likely to be linked with risk of minor or severe IPV (absence of violence group as reference group; see Table 3 for details). Regarding psychological violence, results indicate that while the use of avoidance tended to protect against minor violence (OR = 0.95, *p* < 0.01), participants with an avoidance style were more likely to report severe physical violence victimization (OR = 1.15, *p* < 0.01). Higher self-esteem is associated with a lower risk of minor sexual violence (OR = 0.96, *p* < 0.01), while a rational SPS style is associated with a higher risk. Finally, an ICS is more likely to be linked to the risk of experiencing severe violence (OR = 1.07, *p* < 0.05), especially sexual violence (OR = 1.24, *p* < 0.05).

## 4. Discussion

The identification of life skills that can protect against IPV victimization in emerging adulthood [22,23] is a critical factor in the prevention of interpersonal violence. This French study complements existing European studies on IPV [55,56]. It explores the links between cognitive skills (i.e., SPS), self-esteem and the type and severity of IPV victimization (psychological, physical and sexual violence), and identifies the variables that are most associated with IPV victimization. Our aims were to see whether the severity of psychological, physical and sexual violence is related to low self-esteem and poor SPS skills, and whether negative SPS skills (i.e., NPO, ICS, AS) would be linked to a greater risk of being a victim of these three forms of IPV.

IPV victimization in emerging adulthood presents certain specific dynamics. Descriptive data in this study indicate that psychological violence appears to be the most common form of violence and is concomitant with physical and sexual violence. In terms of gender, the results show that over the past year, 64.5% of the women and 61.6% of the men reported a history of psychological IPV; 16.5% of the women and 18.2% of the men were victims of physical violence; and 22.4% of the women and 9.8% of the men were victims of sexual violence. There is thus no gender difference for psychological and physical victimization, but more women than men are victims of sexual violence. Although there may be controversies in research on the issue of gender symmetry in IPV, the prevalence rates obtained point to a gender symmetry of IPV in emerging adulthood for psychological and physical violence but not for sexual violence [13,14,15,16,17,65].

Different socio-demographic characteristics, such as age, education level and childhood abuse, were identified as significant vulnerability factors for IPV victimization [22,23,37]. These factors are therefore verified in the study and the results do indeed show that a low level of education is associated with more physical violence, and that a history of childhood abuse is associated with more IPV in all its forms.

### 4.1. Self-Esteem, Social Problem Solving and Intimate Partner Violence

General positive SPS skills and good self-esteem levels are associated with lower severity of IPV for all three forms (except for self-esteem and psychological violence). In contrast, negative skills such as having an impulsive/carelessness style (ICS) and an avoidance style (AS) are associated with a greater severity of IPV victimization for all three forms (except AS and sexual violence). As childhood abuse is associated with all three forms of IPV, this may explain why emerging adults have poorer SPS skills. Indeed, being mistreated during childhood can increase exposure to major life stressors, daily problems and severe psychological distress, and may further impair problem-solving skills, which may lead to ineffective problem solving and a greater risk for daily problems and major life stressors in emerging adulthood [37].

The results of the multinomial logistic regressions made it possible to determine whether particular types of SPS skills and self-esteem are more associated with IPV for these three forms. The rational problem-solving style appears to be the skill most associated with minor sexual violence. This is a constructive problem-solving style characterized by rational, systematic, and deliberate use of problem-solving skills. Contrary to expectations, it seems to be a factor of vulnerability, slightly increasing the risk of minor sexual violence. One explanation is that being too rational, conscientious and methodical makes sexual negotiations complicated and can lead to misunderstandings. However, the previous bivariate analyses did not show any significant difference between these variables. Moreover, in the few studies examining these links, no association was found between this style of SPS and sexual violence [57]. This result could be attributed to a lack of robustness of the statistical tests. Low self-esteem also appears to be a factor of vulnerability (with no gender effect) to minor sexual violence. It should be noted that there is a significant gender effect on sexual violence with women being nearly 2.4 times more at risk of being victims of minor sexual violence. This result is consistent with findings that women are more likely to be victims of sexual violence than men [14,17].

An AS appears to be the most explanatory SPS skill for minor psychological violence and severe physical violence. Its style seems to be protective for one and vulnerable for the other. AS is characterized by procrastination, passivity or inaction in SPS. Emerging adults with this style avoid problems rather than confronting them directly, put off solving problems for as long as possible, wait for problems to resolve themselves and attempt to shift the responsibility for solving their problems to others. This type of problem solving to cope with minor psychological violence is used by young adults to guard against violence by avoiding them. Studies investigating AS and victimization of psychological or verbal violence have found a rather positive association [34,36,57]. However, these studies are not nuanced by the severity of the psychological violence, which may explain this result. Conversely, avoiding problems leads to a 1.15-fold increase in the risk of experiencing severe physical violence. Studies by Bell and Higgins and Reich et al. in 2015 also found a positive association between physical violence victimization and AS [37,57]. This avoidance in romantic relationships could bring the other partner to attempt to use physical violence in order to control/establish a hold on him or her [36,54].

ICS seems to be the SPS skill that is most associated with severe psychological or sexual violence. It is characterized by ineffective or inadequate attempts to apply problem-solving skills. Emerging adults with this style do not look for alternative solutions, often impulsively going with the first idea that comes to mind; they scan alternatives and consequences quickly and carelessly and monitor and evaluate solution outcomes inadequately. It seems to be a vulnerability factor, with a 1.07- and 1.24-fold increase in the risk of suffering psychological and sexual violence respectively (bearing in mind that only women are victims of severe sexual violence). Impulsivity is often associated with IPV victimization in the literature [22,66,67]. Bonache et al. also found that engagement in conflict among adolescents could be associated with victimization of psychological or sexual violence [36].

ICS and AS appear to be the main factors of severe forms of IPV victimization (psychological, physical and sexual violence), with ICS leading more to severe psychological and sexual violence, and AS to severe physical violence. We can hypothesize that being subjected to severe forms of physical violence would affect individuals with impulsivity traits less, whereas avoidance, particularly during conflicts, could lead the partner to use physical violence as a means of control. Conversely, impulsivity is a risk factor for perpetrating physical violence [68]. If we consider the bidirectional dynamics of IPV in emerging adults [14,17], we can imagine that if a young person solves problems impulsively, their partner may respond dysfunctionally, using psychological or sexual violence.

### 4.2. Limitations

Several limitations in the current study should be noted. Firstly, as emphasized by Bell and Higgins in 2015 [37], the SPSI-R:SF is not always a suitable way of assessing skills that are generally examined when carrying out a task or exercise. Moreover, it does not define the term “problem”, leaving participants to interpret it in their own way. Likewise, the study did not specifically assess interpersonal SPS as attempted in a previous study [33]. It is possible that participants thought about their abilities to solve non-interpersonal problems when completing the questionnaire, which may be less relevant to the risk of IPV. Secondly, the CTS2 tool can be criticized because it may increase the prevalence rates reported by men or decrease those reported by women and does not take into account the context in which IPV is experienced [62,65]. Future studies could include qualitative designs to be more accurate in assessing IPV. The third limit concerns the design and the sample of the French study. It was a cross-sectional survey, conducted online, and only with a convenience sample of emerging adults, in which women were over-represented. Some IPV severity groups, including severe sexual violence, were smaller in size. Future studies should be conducted with larger, representative samples, and a longitudinal design to better represent all types of IPV, as well as showing changes over time. This would also demonstrate whether SPS and self-esteem predict IPV victimization based on its severity. As suggested by D’Zurilla et al. in 2003 [54], models of mediation between self-esteem and SPS need to be tested to determine whether low self-esteem predicts poor SPS skills, or conversely whether poor SPS skills lead to lower self-esteem, either way leading from vulnerability to IPV victimization. It would also be interesting to conduct longitudinal studies to investigate whether being a victim of IPV induces poor SPS skills and low self-esteem. Fourthly, to better understand the dynamics of IPV in emerging adults, additional studies are needed to investigate victimization and perpetration of IPV to examine not only the bidirectionality but also the polyvictimization and polyperpetration of these forms of violence [69]. Moreover, although the co-occurrence of the three forms of IPV has been described, it has not been integrated into the multivariate analyses so as not to “crush” the associations with the other variables. Latent class analysis or clusters could be proposed to take this co-occurrence into account, resulting in typologies of IPV. Finally, the results must be interpreted with parsimony because the odds ratios of the multinomial logistic regressions remain relatively low, even if they are significant.

### 4.3. Implications and Conclusion

Despite these limitations, this study offers several research contributions and implications. Firstly, it complements the European literature on IPV victimization among emerging adults. Few studies have focused on their links with the resolution of social problems and self-esteem, trying to bring them closer to the concept of life skills. This work therefore constitutes a contribution to research on these dynamics, helps to understand them and helps to develop prevention programs targeted on evidence-based research [24,25,55]. The findings suggest that deficits in SPS skills and low self-esteem may be factors of vulnerability to IPV victimization. This could help set up effective preventive interventions for young adults aimed at improving self-esteem and effective SPS skills. In particular, interventions should focus on overcoming negative problem orientation beliefs and promoting a more positive and optimistic orientation toward problems. Conflict resolution strategies are skills that can be strengthened to prevent IPV [70]. For example, “Safe Dates” [71] or “Fourth R: youth relationships program” [72] are programs that develop these skills and have demonstrated their effectiveness [23,70]. This study suggests the importance of setting up programs to prevent IPV by developing life skills (e.g., problem solving) as early as possible. It also suggests that cognitive-behavioral therapies should be adapted to better target SPS skills that can help individuals find appropriate solutions to cope with their problems [52,73,74,75].

In conclusion, this study extends the existing knowledge reported in European scientific literature. The results support the view that the experience of IPV victimization (psychological, physical and sexual violence) is linked to negative SPS skills, and specifically that ICS, AS and low self-esteem are associated with sexual violence. Identifying the factors that can protect against IPV victimization can help set up preventive interventions. Our findings are preliminary and additional research is needed to further clarify the relationships between SPS, self-esteem and IPV.

## Figures and Tables

**Table 1 behavsci-13-00327-t001:** Gender Differences in Severity of IPV.

	Women *n* = 786		Men *n* = 143		
	*n*	%	*n*	%	*χ* ^2^
Psychological violence					0.57
Absence	279	35.5	55	38.5	
Minor	401	51.0	71	49.7	
Severe	106	13.5	17	11.9	
Physical violence					0.95
Absence	656	83.5	117	81.8	
Minor	95	12.1	21	14.7	
Severe	35	4.5	5	3.5	
Sexual violence					12.25 *
Absence	610 ^b^	77.6	129 ^a^	90.2	
Minor	166 ^b^	21.1	14 ^a^	9.8	
Severe	10	1.3	-	-	

Note. ^a,b^ Frequency differs significantly between men and women. * *p* < 0.05.

**Table 2 behavsci-13-00327-t002:** Differences Between Severity of IPV, SPS and Self-Esteem.

	Psychological Violence					Physical Violence					Sexual Violence				
	Absence 0 *n* = 334	Minor 1 *n* = 472	Severe 2 *n* = 123			Absence 0 *n* = 773	Minor 1 *n* = 116	Severe 2 *n* = 40			Absence 0 *n* = 739	Minor 1 *n* = 180	Severe 2 *n* = 10		
Variables	*M* ± *SD*	*M* ± *SD*	*M* ± *SD*	*F*	*Post-hoc*	*M* ± *SD*	*M* ± *SD*	*M* ± *SD*	*F*	*Post-hoc*	*M* ± *SD*	*M* ± *SD*	*M* ± *SD*	*F*	*Post-hoc*
Age	23.25 ± 3.31	23.80 ± 3.37	23.87 ± 3.36	3.03 *	1 > 0	23.63 ± 3.38	23.44 ± 3.25	23.73 ± 3.19	0.19		23.61 ± 3.35	23.59 ± 3.36	23.80 ± 4.08	0.02	
Education	15.17 ± 2.51	15.03 ± 2.78	14.77 ± 3.07	0.97		15.17 ± 2.68	14.63 ± 2.59	13.83 ± 3.52	6.22 **	0 > 2	15.07 ± 2.68	14.95 ± 2.95	14.50 ± 2.07	0.35	
Childhood abuse	7.81 ± 3.05	8.27 ± 3.24	9.46 ± 3.70	11.62 ***	2 > 1; 2 > 0	8.00 ± 3.07	9.67 ± 3.91	9.18 ± 4.01	15.17 ***	1 > 0	8.08 ± 3.17	8.84 ± 3.53	11.30 ± 4.19	8.43 ***	1 > 0; 2 > 0
PPO	11.50 ± 4.15	11.38 ± 3.87	10.49 ± 4.26	3.01		11.39 ± 4.02	11.06 ± 4.01	10.25 ± 4.32	1.78		11.44 ± 4.05	10.90 ± 3.87	8.20 ± 4.24	4.34 *	2 > 0
NPO	10.97 ± 4.70	10.31 ± 4.62	11.10 ± 4.83	2.19		10.12 ± 4.69	11.27 ± 4.28	11.30 ± 5.32	3.87 *	1 > 0	10.15 ± 4.68	10.93± 4.63	12.70 ± 4.95	3.32 *	1 > 0
RPS	11.37 ± 3.70	11.23 ± 3.75	11.36 ± 3.67	0.16		11.31 ± 3.73	11.51 ± 3.51	10.60 ± 4.02	0.89		11.20 ± 3.71	11.76 ± 3.67	10.40 ± 4.65	2.02	
AS	6.07 ± 4.32	5.51 ± 4.24	7.60 ± 4.59	11.48 ***	2 > 1; 2 > 0	5.72 ± 4.31	6.84 ± 3.66	6.95 ± 4.07	12.10 ***	2 > 1 > 0	5.97 ± 4.38	5.99 ± 4.34	7.10 ± 3.93	0.33	
ICS	5.16 ± 3.63	5.54 ± 3.71	6.38 ± 4.01	4.79 **	2 > 0	5.30 ± 3.71	6.48 ± 3.66	6.95 ± 4.07	8.18 ***	2 > 0; 1 > 0	5.52 ± 3.85	5.24 ± 3.41	9.00 ± 3.37	4.86 **	2 > 0; 2 > 1
Total SPS	61.58 ± 13.94	61.24 ± 14.02	56.76 ± 14.02	6.29 **	2 > 1; 2 > 0	61.58 ± 13.65	57.98 ± 11.71	53.84 ± 14.88	9.08 ***	0 > 1; 0 > 2	60.99 ± 13.70	60.50 ± 12.84	49.80 ± 15.57	3.40 *	0 > 2; 1 > 2
Self-esteem	29.0 ± 6.79	28.40 ± 6.53	27.41 ± 6.61	2.69		28.74 ± 6.64	27.33 ± 6.28	26.85 ± 6.61	3.61 *	0 > 1	28.91 ± 6.54	27.06 ± 6.58	22.6 ± 6.84	9.86 ***	0 > 1; 0 > 2

Note. SPS = social problem solving, NPO = negative problem orientation; ICS = impulsive/carelessness style; RPS = rational problem solving; AS = avoidance style; PPO = positive problem orientation. * *p* < 0.05, ** *p* < 0.01, *** *p* < 0.001.

**Table 3 behavsci-13-00327-t003:** Results of Multinomial Logistic Regressions on Severity of IPV, Self-Esteem and SPS.

	Psychological		Physical		Sexual	
	Minor *n* = 472 OR 95 % CI	Severe *n* = 123 OR 95 % CI	Minor *n* = 116 OR 95 % CI	Severe *n* = 40 OR 95 % CI	Minor *n* = 180 OR 95 % CI	Severe *n* = 10 OR 95 % CI
Variables						
Age	1.07 ** [1.02–1.12]	1.09 * [1.02–1.17]	1.00 [0.94–1.07]	1.06 [0.96–1.16]	1.02 [0.96–1.07]	1.05 [0.86–1.29]
Sex ^a^	1.02 [0.68–1.54]	1.04 [0.75–2.65]	0.79 [.45–1.37]	1.67 [0.60–4.63]	2.38 ** [1.30–4.33]	*-*
Education	0.96 [0.91–1.02]	0.96 [.88–1.04]	0.96 [.89–1.04]	0.87 * [0.77–0.97]	0.98 [0.92–1.05]	1.03 [0.78–1.35]
Childhood abuse	1.04 [0.99–1.09]	1.13 *** [1.06–1.20]	1.13 *** [1.06–1.19]	1.07 [0.97–1.17]	1.06 * [1.00–1.11]	1.17 [1.00–1.38]
Self-Esteem	0.99 [0.96–1.02]	1.00 [0.96–1.04]	1.00 [.96–1.04]	0.99 [0.93–1.05]	0.96 * [0.94–1.00]	0.93 [0.82–1.06]
*SPS*						
PPO	0.99 [0.95–1.05]	0.95 [0.88–1.03]	1.01 [0.95–1.09]	1.00 [0.89–1.12]	0.97 [.92–1.03]	0.81 [0.65–1.02]
RPS	1.03 [0.95–1.05]	1.06 [0.99–1.13]	1.04 [0.97–1.11]	1.06 [0.99–1.13]	1.06 * [1.01–1.12]	1.13 [0.93–1.39]
NPO	1.02 [0.98–1.07]	0.98 [0.92–1.05]	1.03 [0.97–1.10]	0.96 [0.87–1.05]	0.99 [0.94–1.04]	0.95 [0.79–1.14]
AS	0.95 ** [0.91–0.99]	1.05 [1.00–1.11]	1.03 [0.98–1091]	1.15 ** [1.06–1.25]	0.99 [0.95–1.04]	0.94 [1.04–1.11]
ICS	1.03 [0.99–1.08]	1.07 * [1.00–1.14]	1.06 [1.00–1.12]	1.05 [.96–1.15]	0.98 [0.93–1.03]	1.24 * [1.05–1.48]

Note. The reference group was absence of violence; ^a^ the reference group was male; *SPS* = social problem solving, NPO = negative problem orientation; ICS = impulsive/carelessness style; RPS = rational problem solving; AS = avoidance style; PPO = positive problem orientation. * *p* < 0.05, ** *p* < 0.01, *** *p* < 0.001.

## Data Availability

The datasets generated and analyzed during the current study are available from the corresponding author on reasonable request.

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
