# Peer review of "Self-Esteem, Social Problem Solving and Intimate Partner Violence Victimization in Emerging Adulthood"

_behavsci, 2023, doi:10.3390/bs13040327_

Round 1

Reviewer 1 Report

Overall, this paper has a lot of merit and adds to the existing literature in this area.

Introduction/Literature Review

1. There is mention of the 5 features of emerging adulthood, but only 2 are discussed in any detail. A brief sentence or so explaining the other features should be added. (Paragraph 1)

2. Paragraph 3: Lines 68-70. The paragraph is on gendered violence, so one sentence about the negative consequences of IPV seems very out of place here. 

3. There is a lot of discussion about other aspects of IPV, but not a lot about the specific measures examined. More needs to be added about self-esteem and IPV, for example. 

Materials and Methods

4. Intimate partner violence paragraph: what specific items are in the minor and severe IPV scales? Maybe add an appendix 

5. Why are education and history of child abuse being included? Specify the importance of including these variables in the models. 

Results

6. Be careful using 'increased' as that implies causation

Discussion

7. Paragraph 4: Lines 313-314. Child abuse is included in the models, but there's no discussion of it prior to the results/discussion so it is unclear why it is being included. Add some information about the measure and why it is important to include. 

Author Response

We have answered the reviewer, point by point, in the attached file.

Reviewer 2 Report

·       When noting prevalence of IPV for males and females (starts on line 62), are these lifetime prevalence rates? Or within the last year? It is important to note if these are lifetime prevalence rates because of revictimization, which tends to occur at a higher rate once an individual has already been victimized.

·       Since you mentioned witnessing partner violence also increases the risk of experiencing partner violence, one of the main theories associated with an increased risk of IPV is intergenerational transmission of violence, which stems from social learning theory. Is social problem solving also associated with social learning theory? My biggest question for this theory is where are people learning their social problem solving skills? It makes sense the way the authors have it laid out that good problem-solving skills can prevent or reduce IPV, but where are people picking up these skills? Does a model of behavior for problem-solving skills lead to the development of poor problem-solving, subsequently leading to potentially higher rates of IPV? Is the opposite true? A little bit more on the background of theory in regards to where people learn these skills would strengthen the theoretical argument here.

·       What is it about self-esteem that leads to lower or higher risks of IPV? I can infer that it might be since women generally have lower self-esteem that they perceive that IPV is something they “deserve”, and the same might be inferred for men. However, this needs to be expanded upon a little more.

·       One of the biggest issues with the CTS2 is that it produces gender parity effects, i.e., males and females tend to report similar levels of IPV when that isn’t the case – females report more IPV than males (see Hamby, 2014; 2016). Did you encounter this finding when using the CTS2? This should definitely be addressed in the method sections and a follow-up discussion later if you did or did not experience the same issues. This issue should also be discussed in the limitations section.

Author Response

(The authors gave the same response as above.)

Round 2

Reviewer 1 Report

Thank you for making the suggested edits and responding to the comments.

Reviewer 2 Report

Thank you for taking the time to consider my comments and addressing them. I feel the authors did a nice job on expanding their theoretical argument about social problem solving. I also really like what the authors added concerning the CTS2 and the problems with it. Nicely done!